# Weather anomalies more important than climate means in driving insect phenology

R. P. Guralnick [1✉], L. P. Campbell[2] & M. W. Belitz[1]

Studies of long-term trends in phenology often rely on climatic averages or accumulated heat, overlooking climate variability. Here we test the hypothesis that unusual weather conditions are critical in driving adult insect phenology. First, we generate phenological estimates for Lepidoptera (moths and butterflies) across the Eastern USA, and over a 70 year period, using natural history collections data. Next, we assemble a set of predictors, including the number of unusually warm and cold days prior to, and during, the adult flight period. We then use phylogenetically informed linear mixed effects models to evaluate effects of unusual weather events, climate context, species traits, and their interactions on flight onset, offset and duration. We find increasing numbers of both warm and cold days were strong effects, dramatically increasing flight duration. This strong effect on duration is likely driven by differential onset and termination dynamics. For flight onset, impact of unusual climate conditions is dependent on climatic context, but for flight cessation, more unusually cold days always lead to later termination particularly for multivoltine species. These results show that understanding phenological responses under global change must account for unusual weather events, especially given they are predicted to increase in frequency and severity.

[1] Department of Natural History, Florida Museum of Natural History, Dickinson Hall, University of Florida, Gainesville, FL 32611, USA. [2] Florida Medical Entomology Laboratory, Department of Entomology & Nematology, IFAS, University of Florida, 200 9th Street SE, Vero Beach, FL 32962, USA. ✉email: rguralnick@flmnh.ufl.edu

Much of our understanding of long-term trends in phenology comes from examination of either onset or mean timing of events. These studies have shown, across a vast array of lineages and regions, that organisms are shifting growth and reproduction early in the face of current climatic changes (syntheses across animals and plants in refs. [1–3], respectively). Such studies often focus on the yearly or seasonal warming, and look for statistical associations between that warming and trends in key phenological events such as leaf-out[4] or flowering[5] in plants, or timing of developmental phases such as onset of the adult flight period for winged insects[6–9]. While a proliferation of studies continue to show such associations, many continue to rely on yearly or seasonal climatic averages[8]. Even studies that directly use accumulated heat measurements such as growing degree days (GDDs) often aggregate over seasonal or annual periods, rather than directly tying fine temporal grain, e.g. daily environmental conditions, to energetic trade-offs that are highly likely to determine phenology outcomes[10,11].

Unusual warm or cold events may be particularly important as phenology cues[12], and while less predictable than longer-term climate changes, are likely to increase in the future[13]. Schemske et al.[14], for example, demonstrated that unusually warm conditions occurring too early in the season did not trigger understory woodland flowering, but were strongly associated with earlier flowering timing later in the season. Unusually warm spring temperatures resulted in earlier larval hatching of Karner blue butterflies (Lycaeides melissa samuelis), leading to phenological mismatches with its obligate host plant, wild blue lupine (Lupinus perennis)[15]. Unusually warm spring temperatures and early snowmelt resulted in early flight onset of Edith's checkerspot (Euphydryas editha) butterfly, leading to high mortality of adults on at least two occasions over 30 years of a long-term study[16]. In one of these years, high mortality was the result of phenological mismatch with required nectar resources, and in the other the cause was early peak emergence of butterflies that coincided with a large snowfall event[16]. Unusual warming can conversely benefit insect populations by reducing exposure to parasitoids[17,18] or extending length of time for development[19] as has been documented for butterflies in the European Mediterranean region[20]. Drivers of phenology offset, or termination are much less well studied in either plants or insects compared to drivers of onset or peak events. However, the few studies that have examined climate variation, such as ref. [21], have generally found that reduced temperatures increase longevity in cold-tolerant insects, and in particular, that fluctuating thermal regimes delays development and senescence. In summary, intermittent unusual cold may slow metabolic processes without causing lasting damage[21], as long as later normal or warm conditions provide time for recovery.

It may also be that unusual weather events indirectly impact phenology via interactions across trophic levels. It has been shown that excluding pollinators from woodland understory study sites delays individual flower senescence[22], as plants may continue making investment in reproductive structures until pollination occurs. If unusual cold days act to reduce pollinator flights, it may also lead to later flowering senescence, which may also keep phenologies in sync. However, species within and across trophic levels may instead have high variability in onset and termination sensitivity to these anomalous events. If so, such short-term anomalous events could disrupt synchronization of activities, and cascade across trophic levels, as discussed by Butt et al.[23].

Even less is known if anomalous or unusual weather impacts duration of phenophases. If unusual warm events lead to earlier onsets, or unusual cold delays termination, then one or the other should act to increase duration of phenophases. In situations where there are **both** unusual warm and cold events, these may act synergistically to strongly increase duration timing. However, it may ultimately depend on the cadence and intensity of such events and species' physiological tolerances. It may also be that the effect of unusual weather events on phenology are themselves contextually dependent on the overall climatic conditions, such that the impact is weaker or stronger in warmer regions versus colder ones.

To our knowledge, the impact of anomalous events on insect flight onset, termination and duration has not been examined at broad spatial, temporal and phylogenetic scales. Here we use carefully curated phenometric estimates for Lepidoptera (moths and butterflies) of the Eastern USA generated from natural history collections (NHC) data. Lepidoptera are well-suited for phenology studies[24,25], given that they have been well collected for centuries[26], and have temperature-dependent developmental rates[27]. Recent work has also elucidated how key Lepidopteran life-history traits mediate phenological responses, providing a framework for further examining the importance of anomalous weather events. Our focus on using NHC data is intentional, given their immense utility for generating critical phenometrics needed for this work, and unique value for gathering estimates of both phenology onset and termination that span decades and broad spatial extents. This is critical for providing a rich set of estimates of independent unusual weather events, capacity to examine trends over time, and covering broad spatial and environmental gradients.

Based on previous work and given what is already known about butterfly phenological shifts in the face of climate warming, we make four key predictions that we test using our unique dataset. First, we predict that onset occurs earlier when there is preceding unusually warm weather and that this effect is stronger than delayed onset timing from unusual cold. Second, we predict that increasing numbers of unusual cold days will increase delays in adult flight termination. Third, duration of flight periods significantly lengthens in cases where there is both unusually warm and cold weather. Finally, how such unusual weather impacts flight phenology of butterflies and moths is itself determined by both regional climate context and key traits, such as voltinism[28–30].

## Results

We compiled NHC records in order to estimate onset, offset and duration phenometrics. After filtering to the Eastern USA and performing final quality control checks for outliers and removing singleton cells and species, we were left with 850 phenoestimates for 18 unique grid cells covering 139 species, with a temporal span from the 1940s to 2010s. Figure 1 provides a summary of sampling intensity across grid-cells, showing density of estimates, and temporal coverage.

The number of total unusually warm or cold days across all 850 phenoestimates averages was ~2 for both warm and cold ($\bar{x}=2.03$ for warm, $\bar{x}=1.89$ for cold), with a relatively high standard deviation of also ~2 for both. The maximum number of unusually warm or cold days for any phenoestimate was 11 and 13, respectively. The maximum combined sum total of unusual warm and cold days for any phenoestimate was 14. Total number of unusually cold days only modestly positively correlates with GDD for onset ($r = 0.10$) over the same time periods. The correlation between GDD and unusual warm days is even lower ($r = -0.02$). There was a slightly stronger, negative correlation between unusual warm and cold days ($r = -0.23$) but all these values suggest little collinearity.

We ran 2 sets of models, one that used yearly average temperature that included all our phenology estimates, and one that used accumulated GDD over a narrower time window and a

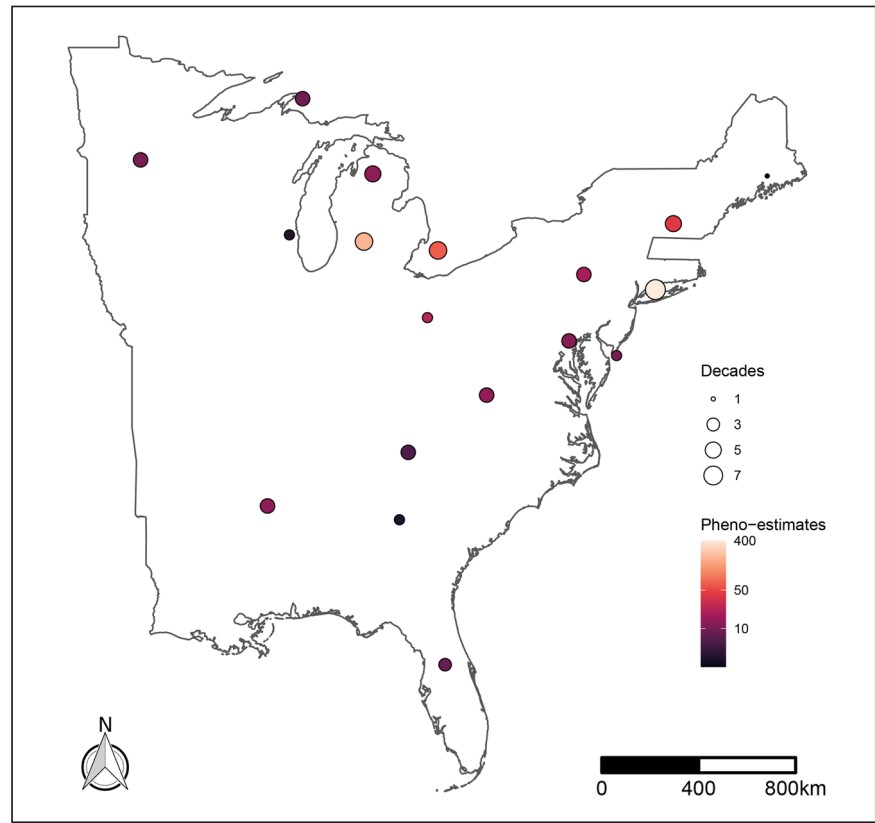

**Fig. 1 Coverage of phenology estimates across the Eastern USA.** Map of coverage and sample size across the Eastern USA. Number of decades covered (size of circles) and total number of phenoestimates (orange to black color ramp) are shown.

subset of earlier flying species. We focus here on results from our annual temperature analyses because the GDD results, while not identical, tell the same story while more limited in the species pool (see Supplemental Figs. 1–5 and Supplemental Table 1). We present results of model fitting for onset and termination first, before turning to duration, with a focus on the impact of unusual cold and warm on phenoestimates. We note that phylogenetic autocorrelation is low across all models based on our PGLMM. We also found little evidence for meaningful temporal or spatial autocorrelation (see Supplemental Figs. 6 and 7) based on residual plots. Here we opt to still present PGLMM model results, which includes information about variance accounted for by random effect terms, including phylogenetic structure, that are retained in model fitting (Table 1).

Model results indicated that a greater number of unusually warm days over a 60-day period prior to onset advances the start of insect flight, while unusually cold days slightly delays the onset of insect flight. However, the effect of unusually cold days is highly dependent on temperature context—in the coldest regions, unusual cold advanced flight onset phenology, while delaying it in the warmest regions (Table 1—onset). The effects of unusual warm days also have a strong association with temperature, with a greater number of unusual warm days in cold regions strongly driving advanced onset but having no effect in the warmest regions (Fig. 2). Based on standardized coefficient estimates, the strongest impacts on onset timing are not climatic, but simply seasonality of flight, voltinism and overwintering stage. However, one trait never included in top models for onset (or any other phenophase) was nocturnal versus diurnal.

Our models for offset showcase an unexpected result. Both unusually cold and warm days act to delay offset, with a stronger effect of unusual cold based on standardized model coefficients.

We also find clear evidence that impacts of unusual warm and cold on termination timing are conditional on climatic context and species traits are also critical, mirroring results for flight onset (Fig. 3A, B). Multivoltine species show much stronger impact of cold days on offset timing, with delayed termination when there is more unusual cold. Univoltine species show much weaker phenological sensitivity to unusual cold, and show more evidence of termination occurring later in colder regions, as opposed to warm regions (Table 1—offset and Fig. 3A). By contrast, greater unusual warm days advance offset in the warmest regions, but delay it in colder areas for both uni- and multivoltine species (Fig. 3B). As with onset models, the strongest predictors of offset timing are trait-based, including season of flight, overwintering stage and voltinism.

Duration models showcase the very strong impact of unusual cold and warm days on insect flight phenology. Unusual cold is a particularly strong driver leading to lengthening flight duration, but as with offset models, unusual warm days also typically lengthens duration. We also find a clear interaction between unusual warm and cold days (Table 1—duration). Figure 4 illustrates the key relationship, showing that unusually cold days drive longer durations, especially in warmer regions. However, when there is both unusual warm and cold, durations lengthen strongly across all climate contexts. As with flight termination timing, the impact of unusual cold on duration is stronger for multivoltine than univoltine species (Table 1).

There are multiple ways to gauge the impact of unusual cold and warm on the duration of flight and here we focus on two approaches. One is to examine rates of phenological change based on model parameters, and the second is to ask how much does model fit decline when key parameters are removed. Based on model parameter estimates, a shift of one standardized unit of

**Table 1 Model parameters for onset, offset, and duration models.**

| Predictors | Onset | Offset | Duration |
|---|---|---|---|
| Intercept | **164.5 (149.3 to 179.5)** | **262.1 (241.9 to 281.9)** | **101.2 (84.8 to 117.4)** |
| Temperature | **−7.6 (−12.5 to −2.7)** | −0.9 (−4.4 to 2.7) | **7.7 (5.8 to 9.7)** |
| Precipitation | 1.1 (−0.9 to 3.2) | **2.4 (0.8 to 4.0)** | 0.5 (−1.1 to 2.1) |
| Unusual cold days | 1.0 (−0.6 to 2.6) | **5.8 (3.8 to 7.7)** | **15.8 (12.7 to 18.8)** |
| Unusual warm days | **−2.0 (−3.4 to −0.6)** | **2.5 (0.8 to 4.2)** | **4.9 (3.3 to 6.5)** |
| Temperature seasonality | **−4.3 (−7.4 to −1.3)** | | |
| Precipitation seasonality | | **−1.7 (−3.1 to −0.4)** | |
| Voltinism [Uni] | −0.6 (−6.8 to 5.5) | **−43.0 (−50.9 to −35.3)** | **−31.1 (−37.9 to −24.4)** |
| Seasonality [Spring] | **−68.1 (−80.8 to −55.2)** | **−55.0 (−73.5 to −36.2)** | 3.8 (−10.4 to 18.1) |
| Seasonality [Summer] | **−29.4 (−37.8 to −21.1)** | **−31.2 (−43.6 to −18.7)** | −0.9 (−10.4 to 8.6) |
| Overwintering strategy [Egg] | **36.0 (20.4 to 51.7)** | 14.7 (−5.3 to 35.0) | **−36.1 (−53.1 to −18.7)** |
| Overwintering strategy [Larvae] | **15.9 (2.5 to 29.2)** | −8.7 (−26.0 to 9.0) | **−31.2 (−45.9 to −16.2)** |
| Overwintering strategy [Migratory] | **25.8 (8.2 to 43.4)** | 15.9 (−8.1 to 39.9) | −15.3 (−34.9 to 4.3) |
| Overwintering strategy [Pupae] | **14.0 (0.3 to 27.7)** | −9.1 (−27.1 to 9.6) | **−30.8 (−45.8 to −15.4)** |
| Temperature: precipitation | **4.3 (2.1 to 6.5)** | **2.6 (0.8 to 4.4)** | |
| Temperature: unusual cold days | **3.6 (1.5 to 5.8)** | | **1.8 (0.3 to 3.3)** |
| Temperature: unusual warm days | 2.1 (−0.1 to 4.2) | −2.6 (−5.4 to 0.1) | **3.1 (0.5 to 5.6)** |
| Temperature: voltinism [Uni] | **−5.1 (−8.7 to −1.4)** | **−8.8 (−12.1 to −5.5)** | |
| Precipitation: unusual cold days | **−2.2 (−3.9 to −0.5)** | | |
| Precipitation: unusual warm days | | 1.3 (−0.3 to 2.9) | **4.6 (2.7 to 6.4)** |
| Unusual cold days: voltinism [Uni] | | **−4.4 (−6.9 to −2.0)** | **−8.5 (−12.0 to −4.9)** |
| Unusual cold days: unusual warm days | | | **4.5 (2.7 to 6.3)** |
| Distinct observation days | **−2.2 (−3.9 to −0.5)** | **4.4 (2.9 to 5.9)** | **5.8 (2.9 to 8.7)** |
| Number of distinct collectors | | | **3.6 (0.8 to 6.4)** |
| Conditional $R^2$ | 0.78 | 0.863 | 0.778 |

Except for Unusual Warm and Cold days, each model contained the same set of predictors. For onset models, we used 60 prior days of unusual warm and cold; for offset, these were measured between onset and offset, and for duration, the sum of the first two. Below we simply refer to "unusual cold days" and "unusual warm days" to simplify presentation of results. Parameters with model coefficients whose 95% Bayesian credible interval does not include zero are in bold. If a model parameter was dropped in the final model or not included, we left parameter estimate value blank.

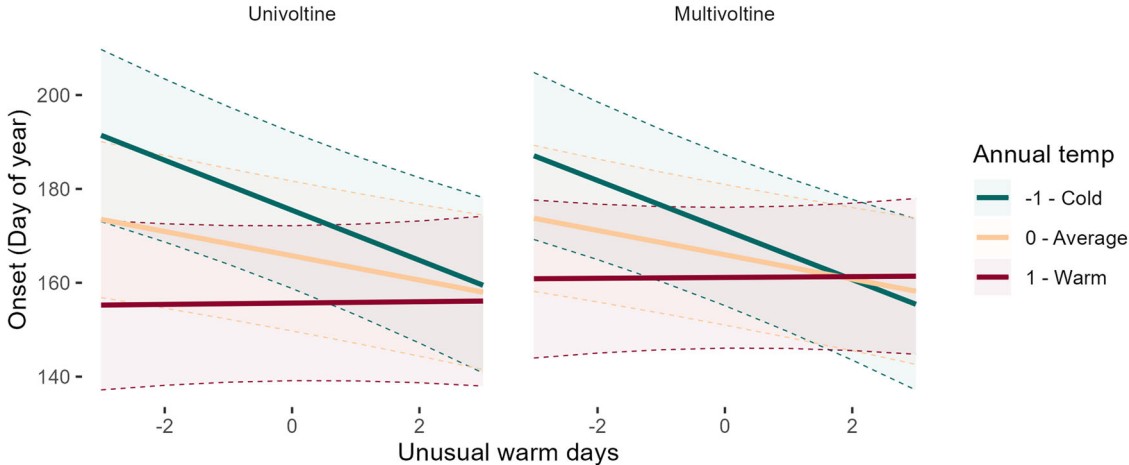

**Fig. 2 Model-based plots of flight start dynamics in relation to unusual warm days and annual temperature.** Effect plots showing flight start dynamics for uni- and multivoltine species in relation to number of warm days and annual temperature. Onset of adult flight period is much earlier when there are more unusually warm days in cold regions, but not in warm regions. While univoltine and multivoltine species have similar sensitivities to unusually warm conditions, onset dynamics across annual temperature gradients are stronger for univoltine species.

both warm and cold days (~6 days total) will lead to ~20 days longer duration of flight. Our model selection procedure using AIC values always found that models that included unusual warm and cold days outcompeted models without unusual warm and cold days as predictors. The conditional AIC of our top linear mixed effect models were considerably lower for models of onset (16.75 ΔAIC), offset (36.91 ΔAIC), and duration (170.03 ΔAIC) when unusual warm and cold days were included in the model.

## Discussion
Recent work on plants (Li et al.[31]) and insects (Belitz et al.[25]) have shown that flowering and insect flight durations are longer in warmer areas in North America. Here we show that this general climate context matters less than unusual weather in determining many aspects of insect adult flight phenology. Lepidopterans, at least, are far more phenologically sensitive to especially unusual cold than to yearly average cold or warm conditions. As an example, for models with annual values, one standardized unit increase in both cold and warm days (~3 days each in unscaled units) will lengthen adult flight duration by nearly 20 days; by contrast, one unit of annual temperature change lengthens duration by only ~8 days (Table 1). We suspect this result extends to insects more generally. Further, the number of unusually warm and cold days are not simply additive, nor are they correlated

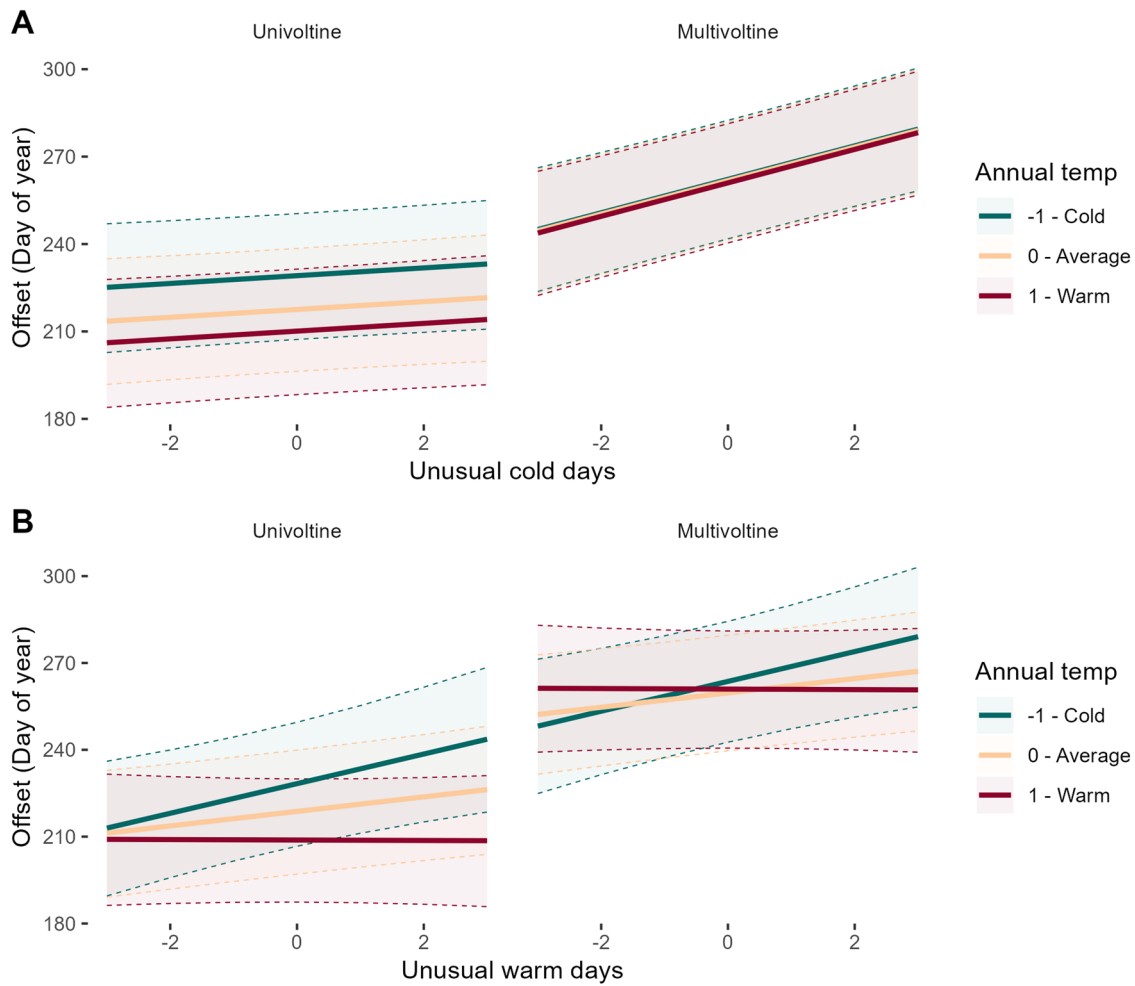

**Fig. 3 Model-based plots of flight termination dynamics in relation to unusual cold days, warm days, and annual temperature.** Effect plots showing termination dynamics for uni- and multivoltine species in relation to number of unusual cold and warm days and annual temperature. **A** Multivoltine species' flight termination timing is more sensitive to unusual cold days than univoltine, but both show longer durations with more unusual cold. As well, multivoltine species flight durations do not differ across climate contexts, but univoltine species have later offsets in colder regions. **B** Unusual warm days in warm contexts slightly advances offset, a result we never find for offset response to unusual cold. Otherwise, in colder contexts, more unusual warm days also delay offset.

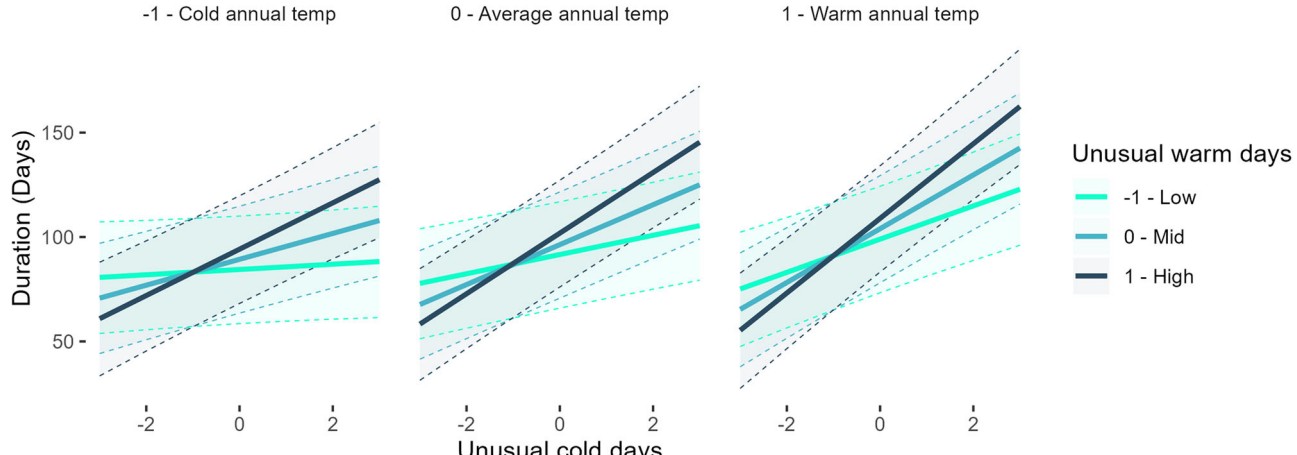

**Fig. 4 Model-based plots of flight duration dynamics in relation to unusual cold days, warm days, and annual temperature.** Effect plots showing duration of flight in relation to number of unusual cold days and warm days in warmer and colder climate contexts. Warm and cold days together interact to increase duration of adult insect flight, with the strongest sensitivity in the warmest regional context. unusual cold days without warm days still increase duration.

strongly with GDDs, a commonly used predictor of phenological responsiveness. We instead found clear evidence of an interaction between the two driving an even stronger response—more of both unusually warm and cold days extend durations longer than either separately.

These results emphatically suggest it is critical to consider unusual or anomalous weather events when understanding shorter or longer-term trends in phenology responses. Such anomalous events are only predicted to be more frequent and severe under climate change[13,32], and thus simply extrapolating average warming as a means to understand phenological change, or potential for mismatch within or across trophic levels, is insufficient. Surprisingly, while unusual weather events have long been suspected to be critical for especially driving phenological mismatch[33], direct empirical demonstration has been limited (but see a within trophic level example in hibernating ground squirrels[34]). Our results clearly demonstrate that incorporating unusual events into frameworks forecasting future phenological changes is essential for informing adaptation strategies under ongoing environmental change, a position that has been advocated by others[35].

The strong effect of unusual cold and warm on duration of flight appears to be driven by differential dynamics and life-history tradeoffs driving flight onset and termination. Lepidopterans in cold regions of the eastern USA are sensitive to unusually warm days prior to flight onset but are not strongly sensitive to unusual cold. In warm regions, it is the reverse, with cold days delaying onset but warm days having little effect. We find similar results for offset, with one critical difference; more unusually cold days always lead to later flight period termination, whether in the warmest conditions or the coldest. In all cases, it is critical to point out that unusually cold and warm conditions are context dependent. That is, we measured unusual cold or warm days against the averages for the region where the phenoestimate was taken; thus unusually cold conditions in northern regions are much colder than unusual cold in southern regions. The differential impact of unusual cold conditions on onset and offset dynamics means that unusual warm days without cold days tend either to lead to static or longer durations, while the converse case of all cold days *always* extends duration via offset dynamics.

Extended durations driven by more weather anomalies was not completely unexpected based on insect thermal biology. Fluctuating thermal conditions, where periods of unusual cold or warm are followed by normal conditions, have been shown to increase developmental time[36] and delay adult senescence[21], although this may be context dependent[27]. These factors may compound in species with multiple clutches per year, as the likelihood for bouts of especially unusual cold during the flight season additively delays production of following generations. We do indeed see a clear effect of stronger duration increases in multivoltine than univoltine species. Still, our approach here does not directly examine the exact timing of unusual weather. Instead, we simply sum the number of unusual days over relevant periods (prior to and during the flight period). The fact that we still see such strong effects simply from summing suggests that even stronger impacts may emerge if variables capturing the cadence and timing of anomalous events were also included.

It is tempting to conclude that unusually warm and cold weather driving longer durations could limit the possibility of mismatches with lower trophic levels, such as plants. However, we note that durations are generally the shortest in warm regions that have a high number of unusual warm days in the absence of unusual cold. As well, our work here is purely focused on phenology of adults, and not earlier life-stage phenology or abundance. Truly extreme events, where temperatures exceed (either positively or negatively) critical survival temperatures for early

larval stages may have enormous impacts on the number of caterpillars who survive to adulthood[37], and on adults themselves[38]. Moving beyond results here therefore requires more careful consideration of across life-stage phenology and abundance.

One challenge with interpreting the growing numbers of phenology studies is choice of climatic summaries related to phenological events[20]. We note that there are a surprisingly limited number of studies that have attempted to examine how weather anomalies impact phenology, and thus no standard method for defining thresholds for "unusual" or "anomalous" conditions. We chose days 2 standard deviations away from long term averages, but one could also consider other ways to capture anomalous events or climate variability (such as coefficient of variation or alternative threshold levels, such as the lower 10th or upper 90th percentile values) over a season.

As well, we found similar results regarding the impact of unusual weather conditions whether using yearly averages and our full dataset or GDDs over a narrower time window focusing on earlier fliers. Still, and unsurprisingly, there are minor differences, such as reduced strength of an interaction of cold and warm days on flight duration. These differences may be due to reduced sample size, effectively removing late fliers (and seasonality as a covariate) in the GDD models. It is critical, therefore, to consider how choices of predictors and filtering of phenology data may matter since these can have consequences for overall interpretation. The choice of climatic context variables may depend on the purpose of the study e.g., better ability to predict phenology versus testing hypotheses or understanding trends. Providing clear and explicit rationale for choices of climate context, along with community coalescence regarding time window choices for climate data assembly, may help especially when comparing across studies and seeking commonalities. Overall, we hope this work motivates more attempts to quantify unusual weather events and their cadence, and how they impact population dynamics, in natural systems. Equally important will be moving from phenomenological studies to experimental ones incorporating climate events[12], in order to fully understand the impact of climate variability on insect phenology and abundance.

We close here focusing on the strong value—and some of the limitations—of using natural history collections data to examine phenological response to climatic variability[26]. On the positive side, NHC data provide a needed basis for generating estimates of the full flight period and extending far back in time. Most other historical datasets are limited to only capturing onset dynamics, while reliance on purely recent data may limit capture of climatic variability, and if weather events are widespread, potentially increase spatiotemporal autocorrelation. On the downside, our estimates are necessarily at relatively coarse spatial grain due to data limitations, and thus our assessment of unusual cold and warm is also over larger regions. This coarse grain also ultimately spatially smooths signals, and thus misses pockets of even more extreme weather conditions. How much incorporating finer-scale phenoestimates and climate data unveil a more nuanced story is an open question. Since climatic data we used here is already fine-grained in both space (800 m × 800 m) and time (daily estimates), the key step towards developing those finer-grained estimates is via increasing density of phenology data. Simulations have shown that our minimum data requirements for fitting phenoestimates are within usable thresholds[39], but even at this coarse spatial resolution, data remains sparse and this has potential to yield imprecise and biased estimates[39]. We encourage continuing, accelerated efforts to digitize insect specimens and linking them to ongoing citizen science collection efforts such as iNaturalist[40].

## Methods

**Determining study area, temporal extent, and assembling phenometrics.** We used phenometric estimates that were generated by ref. [25], but subsetted these to focus on the Eastern United States as defined by the location of the Mississippi River but otherwise including all land area, with the exception of Southern Florida. We chose to focus on the Eastern USA because it is a region where temperature is known to be a strong control on phenology, while in the more arid Western region, precipitation is likely to play a more critical role[41]. While we kept regions bordering Canada, our daily climate data is limited to the conterminous USA. Finally, we limited our temporal extent to 1948–2016 since we opted to use a daily climate dataset with this temporal range, and because using daily data deeper in the past has challenges with increasing, but unknown, uncertainty around climate estimates.

While the details of how phenometrics are estimated are covered in[25], we provide a short summary of the methods used, and note in particular the strong focus on best practices, which is the focus of that work. NHC records were assembled for all North American Lepidopterans from Global Biodiversity Inventory Facility (GBIF), Integrated Digitized Biocollections (iDigBio), and Symbiota Collections of Arthropods Network (SCAN), de-duplicated and filtered for quality as described in ref. [25]. Records passing these filters were joined to a grid of 250 × 250-km equal area cells (North America Albers Equal Area Conic projection). Our choice of this coarse spatial resolution reflects compromises related to data availability of specimen records, but critically, our study focuses on broad-scale patterns across wide gradients that are well-captured at this coarse resolution. Phenometric estimates were generated for species-cell-year combinations where the data were dense enough for usable estimates. We set minimum thresholds separately for univoltine (5 observations, 4 distinct days of collecting, and 3 distinct collectors) and multivoltine (10 total observations, 8 distinct collecting days, and 3 unique collectors) species. These differences reflect the challenges with longer (multivoltine) versus shorter (univoltine) flight periods, based on simulations from ref. [39] and empirical work[42].

Phenometrics for cell-by-year-by-species combinations with enough data were estimated using 5% (hereafter 5% onset or just "onset" or "emergence"), 50%, and 95% (hereafter 95% offset, offset or termination) continuous sample percentiles, along with estimated confidence intervals, using the quantile_ci() function within the "phenesse" R package[39]. Simulations have shown these estimates are relatively robust under low to medium sampling intensity[39] while still capturing a reasonable metric of the bounds of the flight period. Even so, further checks for phenometric outliers were performed by Belitz et al.[25] to remove spurious estimates that may be due to problematic records that cannot be removed via semi-automated means (e.g. dates with transposed day and months, a common issue in digitized specimen records). We added a final check to remove any phenometrics where estimates of duration that were less than 3 days long as likely spurious. Flight duration was calculated as the difference between the 95% termination and 5% onset timing in units of days. Finally, we also assembled distinct days of collecting and number of distinct collectors for each estimate to use downstream as a means to account for sampling effort. We filtered all data to the region of interest (eastern USA) defined above.

**Collating climatic and trait data.** We start by discussing trait data previously collected by[25] in brief. That work assembled two key traits from the literature, "voltinism" and "larval overwintering stage" for each species included in final analyses, which are thought to strongly condition phenology responses. That work also categorized species as "nocturnal and diurnal", which generally follows along phylogenetic lines (diurnal for butterflies, nocturnal for most of the rest of the species). A final key species' trait was flight season, which was categorized into Spring, Summer or Fall by calibrating timing against beginning of the frost-free period, calculating a species mean flight time and binning species by 0–20 (Spring), 20–80 (Summer), and 80–100 (Fall) quantiles, as described in more detail by[25]. Supplemental Table 2 provides a full list of species and traits (also available as a.csv; see data accessibility statement below)[25] also assembled key climatic metrics at the spatial grain of the phenometrics (250 km × 250 km) by aggregating monthly estimated maximum temperature, precipitation data for 1901–2016 at approximately 1-km spatial resolution from the Chelsea data product[43]. The final metrics assembled were yearly precipitation in mm, and mean annual temperature values in degrees C, along with temperature and precipitation seasonality matching bioclimatic definitions (temp seasonality = standard deviation of all months × 100, precip seasonality=coefficient of variation (CV) of monthly precipitation). These were calculated at yearly temporal resolution for all grid cells in the study area. We filtered to the cells and years used in this study.

Key new measurements calculated in this study were unusually or anomalously (we use these terms interchangeably) warm and cold days, defined as days that fell at least 2 standard deviations outside the normal conditions for that day and location when compared to all years used in this study (1948–2016). In order to assemble metrics for unusual warm and cold, we queried the TopoWX dataset[44], an 800-m resolution gridded dataset of daily minimum and maximum air temperature for the conterminous USA. We used the R package "climateR" and the associated package "AOI" to gather daily data over all years for each grid cell where we fit phenometrics and then used raster math to calculate an overall mean temperature per day from 1948–2016 for that cell. We concatenated these data into a single large data table and then wrote two bespoke functions that took estimated onset

and offset day of years and counted how many days prior to onset and between onset and termination of flight were unusually warm (>mean of all years + 2 SD) or cold (<mean of all years − 2 SD). The function allows specification of the number of days prior to onset to examine, and for this study we used 60 days prior. We chose 60 days as a compromise; extending back too far may capture events that are not relevant[14], but too close to onset may miss lag effects. Finally, we used standard "tidyverse" functions[45] to create a sum of unusual warm and cold days for each species-cell-year combination over the period from 60 days prior to onset to flight termination. This approach smooths variation at finer spatial grain and therefore captures spatially broad unusual events rather than localized effects.

We used the same daily data to calculate GDD accumulations in two different ways. First, we were interested in determining the correlations between summed GDD and unusual cold and warm over the same time intervals, to verify lack of strong collinearity. Second, we calculated sum GDD values from March 21 to June 30 for each year and grid cell where we had a phenoestimate, defining the Spring to early Summer period. We then also subsetted our phenometric dataset to only those species, cell, and year combinations where onset was prior to June 30th and determined if our results varied using this predefined window approach compared to using annual temperature. Both GDD measurements used 5 °C as a base minimum threshold and 38 °C as a base maximum threshold. We summed values based on the classic formulation where daily GDD $= [(T_{max} + T_{min})/2] - T_{base\_min}$ (McMaster and Wilhelm 1997)[46] and where if observed $T_{min} < T_{base\_min}$, set to $T_{base\_min}$, and if $T_{max} > T_{base\_max}$ set to $T_{base\_max}$. We chose these relative permissive GDDs, slightly broader than those in ref. [47], given the multispecies approach used here, where species-specific GDD thresholds are not known and the core goal was to compare performance of GDD models and annual temperature models.

**Statistics and reproducibility—modeling butterfly and moth flight onset, termination, and duration.** We initially used a linear mixed modeling approach (LMM) to determine drivers of adult flight onset and termination of phenology, along with duration, while accounting for potential biases in sampling effort. Estimates of the 5% onset, 95% termination, and duration were the response variables for these separate models, and we included mean annual temperature, number of unusually warm and cold days (as separate variables), sum annual precipitation, temperature seasonality, and precipitation seasonality as key climatic predictors. We also included four key traits, flight season, voltinism, overwintering stage and nocturnal vs. diurnal. Finally, we included two predictors capturing sampling effort, number of distinct collecting days and number of distinct collectors. We expected that increased sampling effort over more distinct days by more independent collectors will drive earlier onsets, later terminations and increased durations.

We fit these same models replacing annual temperature metrics with summed GDD values during the narrower window of spring to early summer period. The first set of models using yearly temperature are meant to capture climate context (e.g., warmer years or regions versus colder ones) for unusual weather impacts across all species, whenever they fly. Models incorporating GDD consider accumulation of heat over a time period relevant to the phenology of a subset of species, in this case those that fly earlier in the season, and thus provides a closer link to extrinsic mechanisms impacting phenology. Our goal with fitting both models was to assure that our main results were relatively consistent across approaches and that unusual warm and cold conditions provide unique modeling advantages when using either GDD or annual temperature variables. We carefully checked for collinearity between GDD and unusual warm and cold conditions for both sets of GDD values we assembled (see above).

Before model fitting, we removed grid cells and species with only a single estimate and scaled all non-categorical variables to have a mean of zero and standard deviation (s.d.) of 1 to ensure comparable model effect sizes across variables. Grid cell identity and species were also included as random terms for intercepts but not slopes. Because of the potential for multicollinearity and because our main hypotheses relate to climate interactions and their conditional effects across species with different traits, we focused on a smaller set of two-way interactions between mean annual temperature, sum precipitation, unusual warm and cold days, voltinism and flight season. The unusual warm and cold days sums varied across our three models: for onset models, we used the sums of unusually warm and cold days 60 days prior to onset; for termination models, we used the sums between onset and offset; and for duration we took the overall sum from 60 days prior to onset to termination.

We used the R package "lme4"[48] to fit our LMM models. While we did capture year of phenometric estimate as a variable, we chose to not include this in our models, since we had no direct hypothesis related to year for this work. We did, however, examine potential for temporal autocorrelation post-modeling fitting (Supplementary Fig. 7), as described in more detail below. After initial model fitting, we used the package "lmerTest"[49] to select the best subset model via stepwise variable reduction. We chose lmerTest because it can simultaneously test both fixed and random terms. We re-ran the best reduced models and checked variance inflation using the R package "car"[50]. We found evidence of variance for some interaction terms, removed cases where inflation was detrimental to model parameter estimates (VIF > 5) and then re-fit models. We used the package "performance"[51] to examine pseudo-$R^2$ values for model fits, in particular Nakagawa's $R^2$ [52], designed for mixture models. After fitting models with unusual

warm and cold days included, we also fit a final onset, offset, and duration model where those variables were removed, performing step-wise variable reduction as above, to recover the best model. We then compared the two best models based on Akaike Information Criterion (AIC)[53].

Once we discerned the best models, we used these for a post hoc examination of phylogenetic, spatial and temporal autocorrelation for 2 sets of models; one set with yearly average temperatures and one with GDD. In order to account for phylogenetic autocorrelation, we used an existing tree from[25] subsetted to taxa used just in this study. That tree was built from the Open Tree of Life with branch lengths generated from calibrations from the TimeTree of Life database[54] and scaling of tree using the ph_bladj function from the R package "phylocomr"[55]. The R package "phyr"[56] function"pglmm" was used to fit linear mixed models that incorporate covariance matrices containing the phylogenetic relationships among species as random effects. We used the "bayes" option which fits the models in a Bayesian framework with default uninformative INLA (Integrated Nested Laplace Approximation) priors[57]. We calculated pseudo-$R^2$ values using the method in ref. [25]. We used residuals from the phylogenetic linear mixed model (PGLMM) to examine potential for spatial and temporal autocorrelation. Focusing first on spatial autocorrelation, we calculated Moran's I values across different spatial lags based on a distance matrix of our grid cells. For temporal autocorrelation, we generated autocorrelation function (ACF) plots to examine whether serial correlation in phenology estimates predictably change over time (Supplement 2). Plots of effects and confidence intervals around model estimates were generated from custom code, as described in ref. [9]. All phenometric data and code are available; see data and code availability statements.

**Reporting summary**. Further information on research design is available in the Nature Portfolio Reporting Summary linked to this article.

## Data availability
The data (phenoestimates, traits, climate data, etc.) can be found on GitHub (https://github.com/robgur/LepPheno_UnusualWeather), which is a fork of the Github repository https://github.com/mbelitz/LepPheno_BestPractices, since phenoestimates used here came from that other work. Raw occurrence records needed to replicate our workflow can be downloaded and unzipped from our Open Science Framework project (https://osf.io/wdzay/).

## Code availability
The code to reproduce results and figures presented, including data and scripts, is available on GitHub (https://github.com/robgur/LepPheno_UnusualWeather), which is a fork of the Github repository https://github.com/mbelitz/LepPheno_BestPractices.

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

## Acknowledgements

This work was made possible by the efforts of many, often uncredited museum personnel who collect, accession, preserve, and digitize natural history collections. No funding directly supported this study, but M.W.B. and R.P.G. have been supported to develop methods and tools that enable this work via National Science Foundation DEB # 1703048, including a COVID supplement to this award that proved particularly crucial to allow the time to work on this; L.P.C. is supported under USDA National Institute of Food and Agriculture, Hatch project # 1021482. We also acknowledge the social and economic privileges that have allowed this work, particularly at this time.

## Author contributions

R.P.G., M.B.W., and L.P.C. designed the study. R.P.G. and M.B.W. performed most of the analyses with support from L.P.C. R.P.G. led writing of the manuscript, with help from L.P.C. and M.B.W. All the authors coordinated efforts on revisions and depositing code and data, with R.P.G. leading those tasks.

## Competing interests

The authors declare no competing interests.
