## [Peer Review File · Communications Biology]

Reviewers' comments:

Reviewer #1 (Remarks to the Author):

I carefully read the manuscript "Weather anomalies more important than climate means in driving insect phenology". The manuscript presents a very interesting and new approach to understand how weather relates to butterfly and moth's flight onset, duration and termination including climate variables related to unusual cold and warm daily temperatures. The authors compared different climate predictors on butterfly flight duration and arrive to the conclusion that increasing the number of both warm and cold unusually days is the strongest climate predictor driving longer flight timing. They argued that, in general, unusual cold and warm events (especially the combination of both) are more important in determining insect adult phenology than average climate variables. I enjoyed a lot reading this well-written manuscript with very interesting methods and using an impressive dataset. However, I have some concerns about the analyses that lead the authors to some of their conclusions.

General comments:

The authors compared unusual cold and warm day variables with average climatic variables (temperature and precipitation) to flight onset/offset/duration but these variables are calculated for different time periods. "For onset models, we used 60 prior days of anomalous warm and cold; for offset, these were measured between onset and offset, and for duration, the sum of the first two". However, for average climatic variables they always use the annual mean although annual mean temperatures are not described as a good predictor of insect phenology. In fact, averages temperatures through different time windows importantly change their power to predict butterfly onset (e.g. Gutiérrez and Wilson 2020; Colom et al. 2022). So, I think at least the comparison should account for the same time windows for all the climate variables in the model. I also did not understand why the authors consider GDD to check correlations with the unusual cold and warm days but decided to not include it as an explanatory variable in the models. I'm not sure if the fact that the correlation between GDD and unusually warm and cold days is enough to say that GDD can't capture these anomalous events as the variables proposed by the authors do. As the authors discussed, in this study they chose 2-day standard deviations away from long-term averages to define the unusual cold/warm variables but many other ways of capturing anomalous events can be used. In the case of GDD, it is possible to specify the upper and lower thresholds beyond growing degree days are not accumulating taking into account the physiology of the species based on the information available in the literature. Also, with GDD it is also possible to exploit fine temporal climate data. In this regard, I think GDD can be not only capable to capture anomalous events but also to be able to capture the inter-specific differences in species sensitivity and tolerance to temperature.

Maybe this will not change your results but, to my eyes, it seems a bit arbitrary the selection of 250x250 km cells for the calculations of the butterfly phenology estimates. The authors have available an impressive climate dataset. So, first I would like to see graphically the climate variability in the study area (including the separation between cells). For a European reader these areas seem huge, and I expect high climate variability can be found in some cells (but maybe not). Anyway, I suggest the authors to consider other approaches to select the areas for phenology estimates calculations more based on climate data than on latitude-longitude values. For instance, Schmucki et al. (2016) propose to use bioclimatic regions (e.g. Metzger et al. 2013). Although this it is suggested for standardized count data from butterfly monitoring, I think a similar approach may be applicable for occurrence data. I think it is especially important to consider this because one of the results of your study is the context dependent effect of unusually weather days to flight onset and offset depending on the climate conditions (e.g. cold regions vs warm regions). I also don't know how the authors categorize cells between cold and warm regions (but maybe I missed it).

The authors largely use the reference "Belitz et al. (in press)" to support the analysis used. Then, I think is important that the mentioned paper will be already published by the time of the publication of the present study to be full cited in the references.

Minor comments:

-Page 3, second paragraph: With the examples that the authors provided here, a reader which is not expert in the topic may think that advancing the onset will always have negative demographic consequences on insect species. However, there are some examples that unusual warming in spring may have benefits for insect populations due to the phenological advances driven. For instance, reducing the time that immature stages are exposed to parasitoids (Pollard, 1979; Van Nohuys and Lei, 2004) or increasing the time to complete generations with overall benefits on population abundance of multivoltine insects (Kerr et al. 2020). Indeed, we found that Mediterranean butterflies advancing more their phenology in response to warming have benefits on the overall abundance of their populations compared to the species less phenologically sensitive to temperature (Colom et al. 2022).

-Page 4, last paragraph: "As Belitz et al. (In Press) discuss, Lepidoptera are well-suited for phenology studies, given that they have been well collected for centuries, and have temperature-dependent developmental rates (Buckley, 2022)." Could there be more appropriate references to support this statement?

-Page 4, last paragraph: "Recent work has also elucidated how key Lepidopteran life-history traits..." also require some reference (e.g. Diamond et al. 2011; Zografou et al. 2021, Larsen et al. 2022...).

-Last paragraph of the introduction: Here, you may briefly describe your "unique dataset".

-Page 5, paragraph 3: It may be useful for a non-American reader to get an idea of the coverage of the study if it provides an approximate area in square kilometers.

-Same paragraph: "...in the Western region, precipitation is likely to play a more critical role". I think this sentence also require a reference.

-Page 5, last paragraph: It seems strange that "NHC" is no defined here but in page 16th. You should also provide the full name of "iDigBio".

I would thank if the authors would provide the number of lines in the next version of the manuscript. This would make the review more agile for both parts!

Literature cited:

- Colom, P., Ninyerola, M., Pons, X., Traveset, A., & Stefanescu, C. (2022). Phenological sensitivity and seasonal variability explain climate-driven trends in Mediterranean butterflies. *Proceedings of the Royal Society B*, 289(1973), 20220251.
- Diamond, S. E., Frame, A. M., Martin, R. A., & Buckley, L. B. (2011). Species' traits predict phenological responses to climate change in butterflies. *Ecology*, 92(5), 1005-1012.
- Gutiérrez, D., & Wilson, R. J. (2020). Intra- and interspecific variation in the responses of insect phenology to climate. *Journal of Animal Ecology*, 90(1), 248-259.
- Kerr, N. Z., Wepprich, T., Grevstad, F. S., Dopman, E. B., Chew, F. S., & Crone, E. E. (2020). Developmental trap or demographic bonanza? Opposing consequences of earlier phenology in a changing climate for a multivoltine butterfly. *Global change biology*, 26(4), 2014-2027.
- Larsen, E. A., Belitz, M. W., Guralnick, R. P., & Ries, L. (2022). Consistent trait-temperature interactions drive butterfly phenology in both incidental and survey data. *Scientific reports*, 12(1), 1-10.
- Metzger, M. J., Bunce, R. G., Jongman, R. H., Sayre, R., Trabucco, A., & Zomer, R. (2013). A high-resolution bioclimate map of the world: a unifying framework for global biodiversity research and monitoring. *Global Ecology and Biogeography*, 22(5), 630-638.
- Pollard, E. (1979). Population ecology and change in range of the white admiral butterfly *Ladoga Camilla L.* in England. *Ecological Entomology*, 4(1), 61-74.
- Schmucki, R., Pe'Er, G., Roy, D. B., Stefanescu, C., Van Swaay, C. A., Oliver, T. H., ... & Julliard, R. (2016). A regionally informed abundance index for supporting integrative analyses across butterfly monitoring schemes. *Journal of Applied Ecology*, 53(2), 501-510.
- Van Nohuys, S., & Lei, G. (2004). Parasitoid-host metapopulation dynamics: the causes and consequences of phenological asynchrony. *Journal of Animal Ecology*, 526-535.
- Zografou, K., Swartz, M. T., Adamidis, G. C., Tilden, V. P., McKinney, E. N., & Sewall, B. J. (2021). Species traits affect phenological responses to climate change in a butterfly community. *Scientific reports*, 11(1), 1-14.

Reviewer #2 (Remarks to the Author):

I have included my reviews in the reviewer's document.

** Copied in by editor **

The manuscript "Weather anomalies more important than climate means in driving insect" analyzes the relationship between weather extremes and insect phenologies. The authors use natural history collection archives for calculating insect onset, duration, and termination (phenologies) times and climate data for creating their own defined metric of extreme events. They develop statistical models and supply that with extreme events to explain variabilities in insect phenologies. and finally arrive at a remarkable conclusion: weather extremes are critical in driving inset phenologies. This is an interesting and excellent research paper and has a huge value for the research as well as conservation community not only for understanding the present-day trends but also for the future. I recommend acceptance of the manuscript but with some minor revisions.

You are using multiple species in this study. Different species may overwinter in different stages and in different times of a year. Provide some detail on overwintering stage of different species and some discussion on their overwintering times, this could be a crucial point especially when you are studying multiple species. How variable are the onset, duration and offset times for these different species? Also, data are obtained from the Florida through Minnesota, there is a huge variation in latitude, and there must be variation in onset duration and offset times. Some discussion on this issue would also be helpful. Also, if I understand this correctly, you are using non-migratory species only right? You should mention this.

Abstract

We found increasing numbers of both warm and cold days.... these are the weather events. May be avoid word climate here.

Introduction:

1. First sentence is too long to follow. Break it.
2. Paragraph that starts with Anomalies:

Avoid using "Unusually warm spring" too many times, doesn't read smooth.

3. Paragraph that starts with, Even less is known:

The third sentence: In situation where there are both unusual.... This is very confusing. Are you trying to mean the cold events after the onset event? Or, does it include events even before the onset? I am guessing after the onset, right? Make this clear.

Materials and Methods:

4. First paragraph: We chose to focus on the Eastern USA.... This is a big claim. You need a reference here
5. First paragraph: Since our focus here is on unusual warm and.... narrowing our spatial extent is justified, this is very confusing. Do you mean that there does not exist large variabilities in extreme

precipitation events in the eastern USA? If so, provide a reference. Also, the unusual warm and cold events could also affect the large-scale precipitation that drive local precipitation. I would suggest either to delete or clarify this further. In general, the first paragraph that tries to justify selecting eastern US, east of Mississippi river, is weak. I suggest to provide a few sentences to make it clear and strong.

6. Need a reference in the final sentence of the first paragraph.

7. Paragraph starting with, Phenometrics for cell-by-cell-by-species, include a little more detail on how you define onset and termination here. You mention Belitz paper, but it is a little burden to go find the other paper and get back here.

8. Paragraph starting with, Belitz et al (In Press). Why did you aggregate values in 250km grid? Is there any reason to choose 250km? You have to justify this here. Is it because the locations you obtained NHC data are widely spread across the region?

9. Paragraph starting with, Key new measurements...: When you calculated the extremes, did you also aggregate that to 250km grid? If so, this has to be made clear in this paragraph.

10. Provide explanation on how you accounted for the impact of onset and offset in one grid or location to the neighboring grid or any other grid. I am assuming there is no interaction among the grids, right?

11. Paragraph starting with, Key new measurements..., you mention about how you accumulate the extreme weather days for onset, offset and duration, but that was not very clear until I read heading description of Table 1. You should include those details clearly here also.

12. Why did you use GDD lower threshold 5C? You mention that you did not specify upper threshold, is there any reason for that? Temperatures above 40C (Karlsson and Wiklund 2005) could be lethal and in your case, it would result in higher GDD value. In some location, such lethal temperatures could also be well within 2SD, and may not be counted as extreme in your case. I would suggest to look more into that. You could plot a map of +/- 2 SD temperature values across latitude and include that in supplementary figure or some time series plot since the NHC locations are sparse.

13. Paragraph starting with, Once we discerned the... did you include space and time interaction between adjacent grids in the model? Every location is treated as completely being independent, right ?

Results:

14. Paragraph starting with, "Total number of unusually cold days only modestly positively correlates with GDD ($r=.12$)...." Is this also because you are limiting the lower threshold to 5C and there is no upper threshold. Therefore, it would be interesting to see the number of days that observed temperatures below 5C.

15. Include y-axis [duration (day)] and x-axis label description in the figures 2, 3, 4 as well as their captions. It will be easier to follow by just look at the figures.

Karlsson, B. and Wiklund, C., 2005. Butterfly life history and temperature adaptations; dry open habitats select for increased fecundity and longevity. *Journal of Animal Ecology*, 74(1), pp.99-104.

Reviewer #3 (Remarks to the Author):

General comments

The manuscript uses Lepidoptera natural history collections and gridded climate data in eastern Continental United States to report on how climate affects the onset, termination and duration of adult flight dates. Although a number of climate variables are considered, the hypotheses, data interpretation and discussion focus on the role of the sum of unusual warm and cold days. The manuscript is well-written and was a pleasure to read as a result. My suggestions for improving the manuscript mainly involve the data analyses and interpretation. To frame my suggestions, I point out two potential shortcomings that I see in the data analyses and interpretation.

First, as the introduction points out, much of the past research looking at the role of climate on phenology has used climate means or accumulated growing degree days (GDD). Accumulated GDD are not included as a variable in the mixed model analysis. Instead, their effect is examined in a secondary analysis of the correlation between GDD and the number of warm and cold days. In my view, the correlation creates at best weak inference about the potential effect GDD on phenology. Because accumulated GDD days so often seem to play a role in phenology, here this measure should be treated as null model against which to judge the ability of number of warm and cold days to explain phenology patterns. On a related point, annual mean temperature does appear in the mixed model analyses. However, there was no rationale for the inclusion of this variable after the introduction focused on the importance of climate variability on phenology and discounted, as with accumulated GDD, the importance of climate means for insect phenology.

Second, the linear mixed model analysis was done using a backward selection process that resulted in retention of 12-15 significant variables for the three phenology analyses. Yet, only the variables for the sum of warm and cold days are presented in the results. In two of the three analyses retained variables include statistical interactions involving the sum of warm and cold days, which makes interpreting their main effects problematic. This point is not addressed in the manuscript.

Rather than the backwards selection process that produces one model with a large number of potential confounding variables, I would rather see competing models ranked through AIC. An AIC model selection approach could strengthen the inference by including an explicit null model in the analysis for accumulated GDD. This null model seems especially important to assess because the sum of warm days, for example, could be acting on phenology by bringing accumulated GDD to the critical threshold to set events in motion. A model selection analysis could also result in more parsimonious models so that all important variables could be discussed in the manuscript.

Below, I detail some specific comments. But these will be fewer than normal because the manuscript lacks line and page numbers.

There are several abbreviations in the methods where abbreviations are used without first defining what they mean or are used inconsistently across at different points: p 3 CONUS, p 7. INLA priors, p7 PLMM vs, p 9 PLGMM.

p 7: It appear that two models were built for flight duration, one with warm and cold days and one without these variables. It also appears that a GOF test was used to compare the two models. If so this is the case, GOF tests cannot be used to compare models. There only role is to assess whether a model fits the data. How the GOF is used is unclear because there is no mention of the GOF test in the results and no mention of models fit without warm an cold days in the text or in Table 1.

p.7 Please provide more detail about the how the phylogenetic autocorrelation was included in the

mixed model results in Table 1. This model appears to have been fit with a Bayesian model, yet the only detail about this model given are an abbreviation about some priors, but not which parameters these priors were used for. It also is not clear how the phylogenetic autocorrelation was incorporated into the linear mixed models. Please provide more details about the Bayesian analysis and how its role in the mixed model analyses.

p 8: There are no p-values shown for the correlations, making it difficult to appreciate the result of the analysis.

Figs 2-4: The x-axes in these figures indicates that warm and cold day anomalies were analyzed. This is misleading because these data were the created by summing anomalous cold and warm for each year of the data and standardizing them to mean 0, sd 1. This is not the same as defining an climate anomaly, which must be standardized relative against a run of mean values across a time series to meet that definition. Please change the x-axis figures to note that they are anomalous cold and warm days. Also the error bars are not defined in these figure legends.

These figures also feature discretized presentation of continuous variables (e.g., cold, average and warm temperature in the legend of Fig. 2). But there is no rationale or explanation of why this was done in the methods and how the categories were created. This presentation of the results is a bit confusing as well because temperature was a continuous independent variable in the analysis.

p 10: The model parameter estimates used to estimate phenology change (e.g, for flight duration a ~6 days shift equates to ~20 days longer flight) do not square with the parameter estimates shown in Table 1, so it is not clear how this response was estimated.

Table 1: There a great many significant variables that are retained in the analyses that are not mentioned in the results or discussion. Of particular interest, mean annual temperature is a significant variable in two of the three analyses, but it is mentioned. As importantly, there are significant interactions with the main variables of interest, the number of warm and cold days, that make the main effects difficult to interpret. For example, in the offset analysis there is a cold day x voltinism interaction and in the duration analysis there are temperature x warm day, and temperature x cold day and precipitation x warm day interactions. These interactions should be discussed prominently in the results and discussion.

Dear Editors and Reviewers, we appreciate the very useful reviews of our initial Communications Biology manuscript, “Weather anomalies more important than climate means in driving insect phenology”. We have tried to address, as best we could, all the comments and concerns raised during review. In doing so, we think this is a stronger, sharper and more thoughtful contribution and hope you agree. We provide detailed responses below to reviewers, addressing each issue point by point, starting with Reviewer #1. We’ll look forward to next steps during this process and again we appreciate the time and energy from all involved!

With best regards, Rob Guralnick (on behalf of all the authors).

Reviewers' comments and our Responses

Reviewer #1 (Remarks to the Author):

General comments:

The authors compared unusual cold and warm day variables with average climatic variables (temperature and precipitation) to flight onset/offset/duration but these variables are calculated for different time periods. “For onset models, we used 60 prior days of anomalous warm and cold; for offset, these were measured between onset and offset, and for duration, the sum of the first two”. However, for average climatic variables they always use the annual mean although annual mean temperatures are not described as a good predictor of insect phenology. In fact, average temperatures through different time windows importantly change their power to predict butterfly onset (e.g. Gutiérrez and Wilson 2020; Colom et al. 2022). So, I think at least the comparison should account for the same time windows for all the climate variables in the model.

Response: We agree with this assessment, echoed by other reviewers. We decided the best path here was to define a consistent time window and calculate growing degree days (GDDs). We describe the full details of how we did this below because it is a challenge when working in a multispecies context, where different species have different onset timing. We do want to clarify that we used average annual temperature for three very good reasons in our original manuscript: 1) Annual temperature is routinely used as a measure for studies examining trends over time, and it is very often true that average temperature and GDD strongly correlate; 2) We include many species with different flight periods in a single model in order to look at multispecies responses, and this means picking a GDD window is

challenging; 3) Annual temperature was meant to help provide temperature context in a hypothesis testing framework, rather than prediction.

I also did not understand why the authors consider GDD to check correlations with the unusual cold and warm days but decided to not include it as an explanatory variable in the models. I'm not sure if the fact that the correlation between GDD and unusually warm and cold days is enough to say that GDD can't capture these anomalous events as the variables proposed by the authors do. As the authors discussed, in this study they chose 2-day standard deviations away from long-term averages to define the unusual cold/warm variables but many other ways of capturing anomalous events can be used. In the case of GDD, it is possible to specify the upper and lower thresholds beyond growing degree days are not accumulating taking into account the physiology of the species based on the information available in the literature. Also, with GDD it is also possible to exploit fine temporal climate data. In this regard, I think GDD can be not only capable to capture anomalous events but also to be able to capture the inter-specific differences in species sensitivity and tolerance to temperature.

Response: We opted for a broad rethinking of how to best include GDD and proceeded as follows. First, we honed our focus to Spring and early Summer fliers for the GDD analyses. So, any species by cell by year onset before July 1 was kept (and the rest were filtered out) thus reducing our dataset. This helps us align to a consistent, narrower time window but we still include 65% of the estimates (557 total of 850). We do so because there were no good alternatives for including later season fliers given that GDD rates generally increase in temperate regions towards peak summer and this causes problems with model fitting across too wide a window. We then calculated GDD accumulations in a window from March 21 (Spring Equinox) to June 30th (capturing the Spring period into the earliest Summer). This approach, which includes GDD directly and which narrows to a shared window, is a reasonable test to determine if we find dramatically different results compared to using annual temperature.

Given the reviewer's comment, we also opted to make a shift in our GDD calculation. Here we set the GDD minimum ($T_{basemin}$) to 5C and the maximum at 38F ($T_{basemax}$), given that we include both moths and butterflies in this analysis and it is often the case that multispecies approaches such as the one used here have GDD generic thresholds. Our goal with these generic thresholds was to match what is commonly used in the literature and to avoid thresholds that are too narrow or too wide given known thermal constraints. We do note that these constraints themselves likely vary moderately with latitude, and other factors, but these values provide a reasonable compromise.

We provide a full discussion of the updated methodological details and results in the main text of the paper, the supplements, and in some of the comments below. But here is the short summary: Using this subset analysis and GDDs, we find the same key result -- *unusual warm and cold are highly explanatory in models, more so than GDD accumulation*. Some of the interaction effects are not the same as for the full dataset, but there is no

reason to expect they would be, given that we are using a subset of the data and removed the “season” covariate (since all the species here are spring or early-summer fliers). We also find little evidence of a stronger fit using our subset GDD analysis compared to the full analysis using annual temperature (conditional R^2 are nearly the same). Finally, we re-tested our new GDD values and their correlations with unusual warm and cold days and found generally the same result as before - the correlations were actually slightly weaker and so there is no issue with any harmful covariance.

Given all of this, we decided that the best path forward was to present results from the full dataset, including early and late fliers, using annual temperature as a covariate and provide a supplemental analysis of the subset GDD analysis as a form of sensitivity analysis that provides a useful complement. Given that the supplemental analysis doesn't show a strong interaction between unusual cold and warm impacting duration (as we find for the annual temperature models), we have added a section to the Discussion that discusses some of the challenges with using different sets of predictors in macrophenology studies.

Maybe this will not change your results but, to my eyes, it seems a bit arbitrary the selection of 250x250 km cells for the calculations of the butterfly phenology estimates. The authors have available an impressive climate dataset. So, first I would like to see graphically the climate variability in the study area (including the separation between cells). For a European reader these areas seem huge, and I expect high climate variability can be found in some cells (but maybe not).

Response: The choice of grain here is driven by the observation data we have, and is the same as in Belitz et al. (2021) which uses a different version of the underlying dataset for a very different phenological analysis. The grain size is large but actually slightly smaller than studies by Mayor et al. (2017) and Youngflesh et al. (2020) that examine bird arrival phenology over continental and subcontinental scales. Key here is the focus on a broad extent, where we still capture a wide environmental gradient. With a wide environmental gradient, we are still able to capture significant phenological variation and have reasonable replication over years. In sum, a larger extent doesn't mean that the estimates of either phenology or phenology drivers are inaccurate but it does mean that we are smoothing finer-grain variation. There is strong interest by the authors to further examine some of the questions raised here in additional analyses focused on a multiscale context and that can be accomplished over much shorter time-scales but a much finer grain using resources from community science (rather than a reliance on natural history collections data). Given all this, we don't see a compelling reason to dig deeply into climatic variation within grid cells for this current effort -- surely there is variation -- but one could argue that any grain. Perhaps more interesting is asking how much local scale variation in climate is smoothed moving from the native grain of the climate data to this coarser grain - and whether at this coarser resolution the cold events have to be highly widespread and stronger than similar thresholds at a finer resolution. This all seems like a worthy endeavor for a full study but

doesn't necessarily impact the story here since many macrophenology studies operate at coarse grains.

Anyway, I suggest the authors to consider other approaches to select the areas for phenology estimates based on climate data than on latitude-longitude values. For instance, Schmucki et al. (2016) propose to use bioclimatic regions (e.g. Metzger et al. 2013). Although this it is suggested for standardized count data from butterfly monitoring, I think a similar approach may be applicable for occurrence data. I think it is especially important to consider this because one of the results of your study is the context dependent effect of unusually weather days to flight onset and offset depending on the climate conditions (e.g. cold regions vs warm regions). I also don't know how the authors categorize cells between cold and warm regions (but maybe I missed it).

Response: We appreciate this comment and want to thank the reviewer for a thoughtful review overall. We provide a URL to the now published Belitz et al. (2022) in Functional Ecology (see just below). That paper clarifies how we derive phenoestimates using natural history collections data. We will note that we have used Schmucki's really useful tools for other phenology work but here, the challenge is that estimates for opportunistic data are made by aggregating records within grid cells and then creating a within year temporal series for each species by year by cell combination - that provides a basis for a set of estimators that Belitz et al. (2020) developed and discuss in a paper in Methods in Ecology and Evolution (<https://besjournals.onlinelibrary.wiley.com/doi/10.1111/2041-210X.13448>). Longer story short, our view is that it is absolutely essential to have equal areas for these analyses - broader areas will have earlier onsets and later offsets and longer durations simply because of sampling effort, and while it is potentially possible to account for those biases, we already adjust model calibrations for observer effort and adding more complexity here would only increase uncertainty.

The authors largely use the reference "Belitz et al. (in press)" to support the analysis used. Then, I think is important that the mentioned paper will be already published by the time of the publication of the present study to be full cited in the references.

Response: This paper is indeed published now. Here is the URL:
<https://besjournals.onlinelibrary.wiley.com/doi/abs/10.1111/1365-2435.14173>.

Minor comments:

-Page 3, second paragraph: With the examples that the authors provided here, a reader which is not expert in the topic may think that advancing the onset will always have negative demographic consequences on insect species. However, there are some examples that unusual warming in spring may have benefits for insect populations due to the phenological advances driven. For instance, reducing the time that immature stages are exposed to

parasitoids (Pollard, 1979; Van Nohuys and Lei, 2004) or increasing the time to complete generations with overall benefits on population abundance of multivoltine insects (Kerr et al. 2020). Indeed, we found that Mediterranean butterflies advancing more their phenology in response to warming have benefits on the overall abundance of their populations compared to the species less phenologically sensitive to temperature (Colom et al. 2022).

Response: A great point - we have added in the flip side and cited Colom et al. (2022).

-Page 4, last paragraph: “As Belitz et al. (In Press) discuss, Lepidoptera are well-suited for phenology studies, given that they have been well collected for centuries, and have temperature-dependent developmental rates (Buckley, 2022).” Could there be more appropriate references to support this statement?

Response: Indeed there are some key papers making the same point and we have added another reference e.g. Altermatt et al. (2012).

-Page 4, last paragraph: “Recent work has also elucidated how key Lepidopteran life-history traits...” also require some reference (e.g. Diamond et al. 2011; Zografou et al. 2021, Larsen et al. 2022...).

Response: Added.

-Last paragraph of the introduction: Here, you may briefly describe your “unique dataset”.

We feel this description is better suited for the Methods, and didn't find a logical place where the addition of content about the dataset would help readers “get up to speed”.

-Page 5, paragraph 3: It may be useful for a non-American reader to get an idea of the coverage of the study if it provides an approximate area in square kilometers.

Response: Done.

-Same paragraph: “...in the Western region, precipitation is likely to play a more critical role”. I think this sentence also require a reference.

Response: We added a clarifying statement that much of the Western USA is arid and this is why precipitation is likely an important driver. We also added a reference that follows up this clarification i.e. Currier, C. M., & Sala, O. E. (2022). Precipitation versus temperature as phenology controls in drylands. *Ecology*, 103(11), e3793.

-Page 5, last paragraph: It seems strange that “NHC” is no defined here but in page 16th. You should also provide the full name of “iDigBio”.

Response: Thanks for the useful catch -- an oversight on our part. Now fixed.

I would thank if the authors would provide the number of lines in the next version of the manuscript. This would make the review more agile for both parts!

Response: We agree and this was raised by other reviewers too. Now done!

Literature cited:

Colom, P., Ninyerola, M., Pons, X., Traveset, A., & Stefanescu, C. (2022). Phenological sensitivity and seasonal variability explain climate-driven trends in Mediterranean butterflies. *Proceedings of the Royal Society B*, 289(1973), 20220251.

Diamond, S. E., Frame, A. M., Martin, R. A., & Buckley, L. B. (2011). Species' traits predict phenological responses to climate change in butterflies. *Ecology*, 92(5), 1005-1012.

Gutiérrez, D., & Wilson, R. J. (2020). Intra- and interspecific variation in the responses of insect phenology to climate. *Journal of Animal Ecology*, 90(1), 248-259.

Kerr, N. Z., Wepprich, T., Grevstad, F. S., Dopman, E. B., Chew, F. S., & Crone, E. E. (2020). Developmental trap or demographic bonanza? Opposing consequences of earlier phenology in a changing climate for a multivoltine butterfly. *Global change biology*, 26(4), 2014-2027.

Larsen, E. A., Belitz, M. W., Guralnick, R. P., & Ries, L. (2022). Consistent trait-temperature interactions drive butterfly phenology in both incidental and survey data. *Scientific reports*, 12(1), 1-10.

Metzger, M. J., Bunce, R. G., Jongman, R. H., Sayre, R., Trabucco, A., & Zomer, R. (2013). A high-resolution bioclimate map of the world: a unifying framework for global biodiversity research and monitoring. *Global Ecology and Biogeography*, 22(5), 630-638.

Pollard, E. (1979). Population ecology and change in range of the white admiral butterfly *Ladoga Camilla L.* in England. *Ecological Entomology*, 4(1), 61-74.

Schmucki, R., Pe'Er, G., Roy, D. B., Stefanescu, C., Van Swaay, C. A., Oliver, T. H., ... & Julliard, R. (2016). A regionally informed abundance index for supporting integrative analyses across butterfly monitoring schemes. *Journal of Applied Ecology*, 53(2), 501-510.

Van Nouhuys, S., & Lei, G. (2004). Parasitoid-host metapopulation dynamics: the causes and consequences of phenological asynchrony. *Journal of Animal Ecology*, 526-535.

Zografou, K., Swartz, M. T., Adamidis, G. C., Tilden, V. P., McKinney, E. N., & Sewall, B. J. (2021). Species traits affect phenological responses to climate change in a butterfly community. *Scientific reports*, 11(1), 1-14.

Reviewer #2 (Remarks to the Author):

I have included my reviews in the reviewer's document.

** Copied in by editor **

The manuscript "Weather anomalies more important than climate means in driving insect" analyzes the relationship between weather extremes and insect phenologies. The authors use natural history collection archives for calculating insect onset, duration, and termination (phenologies) times and climate data for creating their own defined metric of extreme events. They develop statistical models and supply that with extreme events to explain variabilities in insect phenologies. and finally arrive at a remarkable conclusion: weather extremes are critical in driving inset phenologies. This is an interesting and excellent research paper and has a huge value for the research as well as conservation community not only for understanding the present-day trends but also for the future.

Response: We appreciate the kind words.

I recommend acceptance of the manuscript but with some minor revisions. You are using multiple species in this study. Different species may overwinter in different stages and in different times of a year. Provide some detail on overwintering stage of different species and some discussion on their overwintering times, this could be a crucial point especially when you are studying multiple species. How variable are the onset, duration and offset times for these different species? Also, data are obtained from the Florida through Minnesota, there is a huge variation in latitude, and there must be variation in onset duration and offset times. Some discussion on this issue would also be helpful. Also, if I understand this correctly, you are using non-migratory species only right? You should mention this.

Response: We thank the reviewer for mentioning overwinter time and stage. Regarding overwintering stage, we have now made sure to provide a separate supplemental file which lists the overwintering stage and diurnality traits for all the species used here. Those are subset from this now-published paper in *Functional Ecology* (see comment for reviewer #1) but we see no harm in a supplemental that is specific to this paper with those traits listed. As for overwintering timing -- we hope the variable "season" that we use in our models (when is the season of flight) is a reasonable proxy for overwintering timing, since we assume that overwintering timing and flight timing are highly correlated. We wish there

were better data to verify just how strong a correlation across many lineages, but since season of flight likely captures the variation you mention, and when diapause breaks is often unknown, we feel we are on solid ground sticking with “season” as a reasonable variable.

Regarding the latitudinal gradient, the reviewer is absolutely correct that we have a significant latitudinal gradient, which is actually essential for us because it also gives us the climatic gradient to see the impacts of temperature on phenology - all of which is accounted for in the model (which incorporates both spatial and temporal components). It is worth noting that what is considered an unusually warm or cold day is context dependent (unusual cold in Florida is not unusual cold in Minnesota!). Finally, in this analysis, we did take on the challenge of including both migratory and resident species in our analysis, and although migratory species had distinctive timing in regards to their onset, offset, and duration of flight periods, a strong interaction between overwintering strategy and climate variables was not retained in any top models.

Abstract

We found increasing numbers of both warm and cold days.... these are the weather events. May be avoid word climate here.

Response: Good point. Fixed.

Introduction:

1. First sentence is too long to follow. Break it.

Response: I don't mind some Steinbeckian versus Hemingwayian sentences but agree and have simplified the sentence.

2. Paragraph that starts with Anomalies:

Avoid using “Unusually warm spring” too many times, doesn't read smooth.

Response: Done

3. Paragraph that starts with, Even less is known:

The third sentence: In situation where there are both unusual.... This is very confusing. Are you trying to mean the cold events after the onset event? Or, does it include events even before the onset? I am guessing after the onset, right? Make this clear.

Response: Clarified (as best we can).

Materials and Methods:

4. First paragraph: We chose to focus on the Eastern USA.... This is a big claim. You need a reference here

Response: Agree and raised in other reviews too. See response above.

5. First paragraph: Since our focus here is on unusual warm and.... narrowing our spatial extent is justified, this is very confusing. Do you mean that there does not exist large variabilities in extreme precipitation events in the eastern USA? If so, provide a reference. Also, the unusual warm and cold events could also affect the large-scale precipitation that drive local precipitation. I would suggest either to delete or clarify this further. In general, the first paragraph that tries to justify selecting eastern US, east of Mississippi river, is weak. I suggest to provide a few sentences to make it clear and strong.

Response: We think adding in clearly that we chose to avoid regions where aridity is a known phenology driver (along with a recent reference) and this would exclude much of the Western USA is helpful here. The additions are minimal but clear and we opted to delete the sentence the reviewer suggested. This now reads much clearer and stronger to us (and hopefully the reviewer agrees).

6. Need a reference in the final sentence of the first paragraph.

Response: Done (also requested by reviewer 1).

7. Paragraph starting with, Phenometrics for cell-by-cell-by-species, include a little more detail on how you define onset and termination here. You mention Belitz paper, but it is a little burden to go find the other paper and get back here.

Response: The next paragraph has a pretty clear statement we tried to further improve. Now: "Phenometrics for cell-by-year-by-species combinations with enough data were estimated using 5% (hereafter 5% onset or just "onset" or "emergence"), 50%, and 95% (hereafter 95% offset, offset or or termination) continuous sample quantiles, along with estimated confidence intervals, using the `quantile_ci()` function within the "phenesse" R package (Belitz et al., 2020). Simulations have shown these estimates are relatively robust under low to medium sampling intensity (Belitz et al., 2020) while still capturing a reasonable metric of the bounds of the flight period."

8. Paragraph starting with, Belitz et al (In Press). Why did you aggregate values in 250km grid? Is there any reason to choose 250km? You have to justify this here. Is it because the locations you obtained NHC data are widely spread across the region?

Response: We discuss this above as Reviewer 1 had a similar comment - we hope the explanation is clear and have made text edits that hopefully communicate the methods more clearly while refraining from overwhelming technical detail on how to generate phenometrics using incidental records.

9. Paragraph starting with, Key new measurements...: When you calculated the extremes, did you also aggregate that to 250km grid? If so, this has to be made clear in this paragraph.

Response: Good point - the extremes are based on the aggregated climate data to the 250X250km extremes. This means that the region encompassed by the grid cell, give or take, had to basically be in an extreme event and gets into some of the complexity of scale issues here. We have added in this clarification i.e. at the end of that paragraph, "This approach smooths variation at finer spatial grain and therefore captures spatially broad unusual events rather than localized effects."

10. Provide explanation on how you accounted for the impact of onset and offset in one grid or location to the neighboring grid or any other grid. I am assuming there is no interaction among the grids, right?

Response: We did look at spatial autocorrelation directly and did not detect significant residual autocorrelation - see the supplementals (especially Supp. 1).

11. Paragraph starting with, Key new measurements..., you mention about how you accumulate the extreme weather days for onset, offset and duration, but that was not very clear until I read heading description of Table 1. You should include those details clearly here also.

Response: Fair point and we have tried to make some edits for clarity here.

12. Why did you use GDD lower threshold 5C? You mention that you did not specify upper threshold, is there any reason for that? Temperatures above 40C (Karlsson and Wiklund 2005) could be lethal and in your case, it would result in higher GDD value. In some location, such lethal temperatures could also be well within 2SD, and may not be counted as extreme in your case. I would suggest to look more into that. You could plot a map of +/- 2 SD temperature values across latitude and include that in supplementary figure or some time series plot since the NHC locations are sparse.

Response: We used 5C as a generic threshold and we do agree that extremely high temperatures might be lethal -- but it is very difficult to actually set a reasonable expectation in a multispecies study where species vary by latitude and season --- they are likely adapted to different ranges of temperatures making choices here tricky. In the end, we opted to go with an upper threshold of 38F as capturing what the reviewer noted above.

We believe these generic thresholds likely capture a reasonable snapshot or “average” of the thermal ecology that is reasonable given the multispecies nature of the study and we provide in text citations to other work that use similar (albeit somewhat more restrictive values). We note that one good thing about our conceptualization of extreme warm and cold is that it is relative to average conditions at a cell and time period within the year, which might help to mitigate using the same thresholds on the GDD side for all species.

13. Paragraph starting with, Once we discerned the... did you include space and time interaction between adjacent grids in the model? Every location is treated as completely being independent, right ?

Response: Yes we have independently assessed values for each grid by year by species, and then checked for both spatial and temporal autocorrelations and their importance.

Results:

14. Paragraph starting with, “Total number of unusually cold days only modestly positively correlates with GDD ($r=.12$)....” Is this also because you are limiting the lower threshold to 5C and there is no upper threshold. Therefore, it would be interesting to see the number of days that observed temperatures below 5C.

Response: We agree that correlations might differ because of the way we calculated GDD. We re-did these correlations for the GDD supplemental analysis and found effectively the same results (low or no correlation) using a different conception of GDDs (a single window versus looking at GDD over the same period as the unusual warm and cold days).

15. Include y-axis [duration (day)] and x-axis label description in the figures 2, 3, 4 as well as their captions. It will be easier to follow by just look at the figures.

Response: We have completely redone figures and captions based on the comments from reviewers - this should help with readability more generally.

Karlsson, B. and Wiklund, C., 2005. Butterfly life history and temperature adaptations; dry open habitats select for increased fecundity and longevity. *Journal of Animal Ecology*, 74(1), pp.99-104.

Reviewer #3 (Remarks to the Author):

General comments

The manuscript uses Lepidoptera natural history collections and gridded climate data in eastern Continental United States to report on how climate affects the onset, termination and duration of adult flight dates. Although a number of climate variables are considered,

the hypotheses, data interpretation and discussion focus on the role of the sum of unusual warm and cold days. The manuscript is well-written and was a pleasure to read as a result. My suggestions for improving the manuscript mainly involve the data analyses and interpretation. To frame my suggestions, I point out two potential shortcomings that I see in the data analyses and interpretation.

First, as the introduction points out, much of the past research looking at the role of climate on phenology has used climate means or accumulated growing degree days (GDD). Accumulated GDD are not included as a variable in the mixed model analysis. Instead, their effect is examined in a secondary analysis of the correlation between GDD and the number of warm and cold days. In my view, the correlation creates at best weak inference about the potential effect GDD on phenology. Because accumulated GDD days so often seem to play a role in phenology, here this measure should be treated as null model against which to judge the ability of number of warm and cold days to explain phenology patterns. On a related point, annual mean temperature does appear in the mixed model analyses. However, there was no rationale for the inclusion of this variable after the introduction focused on the importance of climate variability on phenology and discounted, as with accumulated GDD, the importance of climate means for insect phenology.

Response: This important and clearly stated point strongly echoes a key point from Reviewer #1, and we refer to the response to Reviewer #1 here and what has been done regarding re-running models to include GDD as a variable. The models we ran explicitly test what happens when we control for GDD (or mean annual temperature) and look at the effect of unusual warm and cold, and we basically show the same key result for both. As a side note: what we cannot do is use the same window of GDD as for unusual warm or cold. This is because unlike our cold and warm “deviations” which are relative across seasons and regions, GDD accumulations are sums. The end result when one tries this are nonsense models that show higher GDD incorrectly leading to later, not earlier, onset because late fliers are closer to peak GDD accumulation in mid-summer. The only way to do these GDD analyses is to pick windows where climate and phenology events align.

Second, the linear mixed model analysis was done using a backward selection process that resulted in retention of 12-15 significant variables for the three phenology analyses. Yet, only the variables for the sum of warm and cold days are presented in the results. In two of the three analyses retained variables include statistical interactions involving the sum of warm and cold days, which makes interpreting their main effects problematic. This point is not addressed in the manuscript. Rather than the backwards selection process that produces one model with a large number of potential confounding variables, I would rather see competing models ranked through AIC. An AIC model selection approach could strengthen the inference by including an explicit null model in the analysis for accumulated GDD. This null model seems especially important to assess because the sum of warm

days, for example, could be acting on phenology by bringing accumulated GDD to the critical threshold to set events in motion. A model selection analysis could also result in more parsimonious models so that all important variables could be discussed in the manuscript.

Response: The challenge we face is that whether one does forward or backward model selection, we have a large number of covariates and potential interactions, and we are not fans of simply dredging 100s of competing models for onset, offset and duration. Since our key question in this manuscript is how unusually warm and cold days impact model fits, we opt for starting with full models for onset, offset and duration as discussed below and examining the AIC of competing top models after running a backward selection approach on these 'global' models.

We use duration models as our example because they are focal for this paper. The first two full candidate models focus on annual temperature: one is a full model with annual temperature, along with unusual warm and cold days, plus all the other fixed and random effects, and one that just uses annual temperature (plus other random+fixed effects). We opt to still use stepwise backward selection using `step()` in `lmerTest` and checks of variance inflation using the `vif()` function in the R package "car" in order to fit a best model without harmful collinearity. However, we explicitly evaluate the conditional AICs of the models to determine the lowest AIC between the two.

We do the same process separately for the GDD models. Because the GDD models use a subset of the data, we can't directly compare GDD and annual temperature models using AICs. We want to be clear as well that it is the best model from this process that is then further evaluated for spatial, temporal and phylogenetic autocorrelations. This is another compromise since trying to fit all those via model selection would even further (and unnecessarily, in our view) increase complexity.

While we understand the rationale for considering something akin to a "null" model (e.g. intercept-only or annual temperature as a single covariate), we know unequivocally here that both the random effects and many of the fixed effects such as season or observer effort are critical for properly fitting these models - this has been shown repeatedly in empirical studies and we explicitly mention its importance as a best practice in our Functional Ecology paper (Belitz et al., 2022). We think this approach, which focuses on the key hypotheses of interest, is a useful compromise between dredging and too much handpicking models to evaluate. As for the key result: *We always find that models including unusual warm and cold are better than models that don't, whether using the full dataset or using a subset for our GDD-based analyses.*

Below, I detail some specific comments. But these will be fewer than normal because the manuscript lacks line and page numbers.

There are several abbreviations in the methods where abbreviations are used without first defining what they mean or are used inconsistently across at different points: p 3 CONUS, p 7. INLA priors, p7 PLMM vs, p 9 PLGMM.

Response: We now spell these out - good point!

p 7: It appear that two models were built for flight duration, one with warm and cold days and one without these variables. It also appears that a GOF test was used to compare the two models. If so this is the case, GOF tests cannot be used to compare models. There only role is to assess whether a model fits the data. How the GOF is used is unclear because there is no mention of the GOF test in the results and no mention of models fit without warm an cold days in the text or in Table 1.

Response: A fair point and we have proceeded with rankings using AIC here as a key means to compare models, as described in more detail above.

p.7 Please provide more detail about the how the phylogenetic autocorrelation was included in the mixed model results in Table 1. This model appears to have been fit with a Bayesian model, yet the only detail about this model given are an abbreviation about some priors, but not which parameters these priors were used for. It also is not clear how the phylogenetic autocorrelation was incorporated into the linear mixed models. Please provide more details about the Bayesian analysis and how its role in the mixed model analyses.

Response: This is our fault -- we use this approach fairly regularly and in this our oversight stopped us from providing as much detail as would be useful for readers. We have added text for clarity on this approach but the exact Bayesian implementation is beyond the scope of this contribution and we refer the reviewer to the “phyr” R package and associated application paper for more details.

p 8: There are no p-values shown for the correlations, making it difficult to appreciate the result of the analysis.

Response: We are not huge fans of p-values, since with very large samples, p-values can be very high but capture incrementally small differences - we opt to not include them and focus instead on variance explained. As well, since we now include models where GDD is directly included as a covariate, we can test for collinearity using variance inflation factors, and we never see any evidence of significant issues with collinearity between GDD and unusual warm and cold days.

Figs 2-4: The x-axes in these figures indicates that warm and cold day anomalies were analyzed. This is misleading because these data were the created by summing anomalous cold and warm for each year of the data and standardizing them to mean 0, sd 1. This is not

the same as defining an climate anomaly, which must be standardized relative against a run of mean values across a time series to meet that definition. Please change the x-axis figures to note that they are anomalous cold and warm days. Also the error bars are not defined in these figure legends.

Response: A great point. We fixed this by making legends and axes more clear.

These figures also feature discretized presentation of continuous variables (e.g., cold, average and warm temperature in the legend of Fig. 2). But there is no rationale or explanation of why this was done in the methods and how the categories were created. This presentation of the results is a bit confusing as well because temperature was a continuous independent variable in the analysis.

Response: This is simply a visualization choice that we used for plotting model results that would be easier for readers to follow, rather than any discretization or binning prior to model runs. Basically, once you exceed two variables for plotting predictions along the x and y, the only choice for showing another variable's impact is to either go for a 3D plot or to show a prediction across a range of values. Here we choose to show a prediction across three annual temperature ranges: cold (which corresponds to an area 1 SD below the mean), average (corresponding to an area with mean annual temperature), and warm (an area 1 SD above the mean). We have made the figure legends more clear here and we think this likely helps allay concerns raised above without having to add in more to the figure caption.

p 10: The model parameter estimates used to estimate phenology change (e.g, for flight duration a ~6 days shift equates to ~20 days longer flight) do not square with the parameter estimates shown in Table 1, so it is not clear how this response was estimated.

Response: Good this was checked. We have gone back and verified these numbers. We didn't explain this particularly well, but # of unusual cold days leads to 16 days longer and # of unusual warm days leads to 5 longer, which together sums to 21 days longer. We made some adjustments to wording, and appreciate diligence in checking our math.

Table 1: There are a great many significant variables that are retained in the analyses that are not mentioned in the results or discussion. Of particular interest, mean annual temperature is a significant variable in two of the three analyses, but it is mentioned. As importantly, there are significant interactions with the main variables of interest, the number of warm and cold days, that make the main effects difficult to interpret. For example, in the offset analysis there is a cold day x voltinism interaction and in the duration analysis there are temperature x warm day, and temperature x cold day and precipitation x warm day interactions. These interactions should be discussed prominently in the results and discussion.

Response: We agree, to a degree, with this comment. We opted in this paper to try to keep our focus as much as we possibly could on the hypotheses that we felt were open for testing -- how do weather anomalies impact phenology. The challenge is that macrophenology studies can and should try to account for as much variability in life history traits and other forcing mechanisms as possible to derive generalizable results. In many ways, the most logical next step once we have isolated a pattern or association that we think is real, is to build an experimental approach where we have more control. We have added in what amounts to a new paragraph that brings up some of the challenges and complexities in macrophenology studies that dovetails with the issue(s) raised by the reviewer (the one starting with "As well, we found similar results regarding the impact of unusual weather conditions whether using yearly averages and our full dataset, or growing degree days over a narrower time window focusing on earlier fliers."). Still, we have otherwise tried to avoid some weeds for the sake of clarity and staying on message.

REVIEWERS' COMMENTS:

Reviewer #2 (Remarks to the Author):

To Guralnick et al.

I read all your responses. Thank you for carefully addressing my concerns.

Congratulations!

Thank you so much!

Reviewer #3 (Remarks to the Author):

The authors thoroughly replied to all substantive reviewer comments and have done a good job revising the manuscript.